# Broccoli, Amaranth, and Red Beet Microgreen Juices: The Influence of Cold-Pressing on the Phytochemical Composition and the Antioxidant and Sensory Properties

**DOI:** 10.3390/foods13050757

**Published:** 2024-02-29

**Authors:** Spasoje D. Belošević, Danijel D. Milinčić, Uroš M. Gašić, Aleksandar Ž. Kostić, Ana S. Salević-Jelić, Jovana M. Marković, Verica B. Đorđević, Steva M. Lević, Mirjana B. Pešić, Viktor A. Nedović

**Affiliations:** 1Food Biotechnology Laboratory, Department of Food Technology and Biochemistry, Faculty of Agriculture, University of Belgrade, Nemanjina 6, 11080 Belgrade, Serbia; sbelosevic@agrif.bg.ac.rs (S.D.B.); ana.salevic@agrif.bg.ac.rs (A.S.S.-J.); jovana.markovic@agrif.bg.ac.rs (J.M.M.); slevic@agrif.bg.ac.rs (S.M.L.); 2Food Chemistry and Biochemistry Laboratory, Department of Food Technology and Biochemistry, Faculty of Agriculture, University of Belgrade, Nemanjina 6, 11080 Belgrade, Serbia; danijel.milincic@agrif.bg.ac.rs (D.D.M.); akostic@agrif.bg.ac.rs (A.Ž.K.); 3Department of Plant Physiology, Institute for Biological Research Siniša Stanković-National Institute of Serbia, University of Belgrade, Bulevar Despota Stefana 142, 11060 Belgrade, Serbia; uros.gasic@ibiss.bg.ac.rs; 4Department of Chemical Engineering, Faculty of Technology and Metallurgy, University of Belgrade, Karnegijeva 4, 11000 Belgrade, Serbia; vmanojlovic@tmf.bg.ac.rs

**Keywords:** broccoli microgreens, amaranth microgreens, red beet microgreens, microgreen juices, antioxidant activity, apigenin *C*-glycosides

## Abstract

The aim of this study was to analyze in detail the phytochemical composition of amaranth (AMJ), red beet (RBJ), and broccoli (BCJ) microgreens and cold-pressed juices and to evaluate the antioxidant and sensory properties of the juices. The results showed the presence of various phenolic compounds in all samples, namely betalains in amaranth and red beet microgreens, while glucosinolates were only detected in broccoli microgreens. Phenolic acids and derivatives dominated in amaranth and broccoli microgreens, while apigenin *C*-glycosides were most abundant in red beet microgreens. Cold-pressing of microgreens into juice significantly altered the profiles of bioactive compounds. Various isothiocyanates were detected in BCJ, while more phenolic acid aglycones and their derivatives with organic acids (quinic acid and malic acid) were identified in all juices. Microgreen juices exhibited good antioxidant properties, especially ABTS^•+^ scavenging activity and ferric reducing antioxidant power. Microgreen juices had mild acidity, low sugar content, and good sensory acceptability and quality with the typical flavors of the respective microgreen species. Cold-pressed microgreen juices from AMJ, RBJ, and BCJ represent a rich source of bioactive compounds and can be characterized as novel functional products.

## 1. Introduction

Microgreens are recognized as new crops and potential foods of the future [1]. They represent a novel and promising source of highly valuable bioactive compounds with health-promoting effects [2,3,4,5,6]. The most commonly grown and studied microgreens are from the Brassicaceae and Amaranthaceae families with crops such as broccoli, cabbage, kale, argula, red beet, chard, amaranth, etc. [1]. So far, the aforementioned microgreen species have been mostly consumed in raw form or as culinary ingredients in dishes due to their high content of bioactive compounds and specific flavor [7]. Previous studies have shown that broccoli, amaranth and red beet microgreens are high in bioactive compounds such as vitamins, glucosinolates, isothiocyanates, phenolic compounds and betalains and have good antioxidant properties [4,8]. However, there are few studies that provide insight into the biocompound profiles of these microgreens and their correlation with antioxidant properties [9,10,11,12,13,14]. In addition, microgreens can be successfully used for healthy beverage production [15,16,17,18] or incorporated into various bakery/confectionary products [19]. Some studies have preliminarily analyzed microgreen juices from broccoli [15], Alternanthera sessilis [16], and wheatgrass [17], as well as functional microgreen/fruit juices [20]. However, in general, microgreen juices have only become attractive in recent years and have not been extensively studied. Various processed, treated, or stored wheatgrass and wheat sprout juices have shown good scavenging activity of ABTS^+^, DPPH, and oxygen radicals [17,18,21]. These antioxidant assays have not yet been performed on juices from other microgreens. Recent and rare in vitro and in vivo studies conducted with cold-pressed broccoli microgreen and sprout juices have shown that the juices may have anticancer [22] and antiobesity effects [15] and protective properties against oxidative stress-related diseases [23]. However, considering previous research on microgreen juices [15], their health effects, and the lack of their bioactive compounds profile, further research is thus needed. On the other hand, microgreen juices from amaranth and red beet have not been analyzed to our knowledge. Finally, sensory analysis is often a key parameter for the acceptance of new products based on microgreens, mainly due to their specific flavor attributes. Tests for consumers’ sensory perception and acceptance of microgreens have been conducted most frequently [24,25,26,27,28], while microgreen juices have hardly been tested. To date, only microgreen juices of Alternanthera sessilis and Brassica juncea have been the subject of sensory evaluation [16], showing good overall acceptability, using the hedonic test. To our knowledge, sensory evaluations have not yet been performed on the juices from broccoli, amaranth, and red beet microgreens. However, these microgreen varieties have unique sensory attributes such as astringency, bitterness, and sourness which most likely contribute to the overall acceptability of their products by consumers [24,25].

In view of the aforementioned, the aim of this study was to prepare cold-pressed juices from broccoli, amaranth, and red beet microgreens and to analyze in detail their phytochemical composition, antioxidant properties, and sensory acceptability. In addition, the profiles of bioactive compounds of raw broccoli, amaranth, and red beet microgreens were also analyzed to better follow the migration of individual glucosinolates, phenolic compounds, and betalains from the microgreens to juices and to explain their transformation during the juices’ production process.

## 2. Materials and Methods

### 2.1. Microgreen Sample

Samples of broccoli (*Brassica oleracea* var. *italica*), red beet (*Beta vulgaris*), and amaranth (*Amaranthus tricolour* L.) microgreens were obtained from a local company (Plantica) from Belgrade, Serbia. Briefly, the microgreens were grown in a controlled environment, including vertical cultivation in the growing channels. They were grown under artificial light and at room temperature (20 °C). The humidity in the room (85%) and the air temperature was ensured by fans. The year in which the microgreens of broccoli, amaranth, and red beet were produced was 2023. All microgreens used in this study were harvested 12 days after germination and when the first pair of true leaves and the fully expanded embryonic leaves (cotyledons) had developed.

### 2.2. Preparation of Cold-Pressed Microgreen Juices

Selected microgreens were cut with scissors a few centimeters above the ground, then weighed and washed to remove impurities. The cleaned microgreens were pressed using a super-slow juicer (Angel juicer 8500, Angel Co., Ltd., Busan, Republic of Korea), and the obtained juices from broccoli (BCJ), amaranth (AMJ), and red beet (RBJ) (cold-pressed microgreen juices) were collected in plastic flasks (Figure 1). Part of the prepared juices was stored in the refrigerator for sensory evaluation. The other part of the squeezed juices was centrifuged at 9000× *g* rpm for 12 min to remove solid fractions, and the collected supernatants were stored at −20 °C for further spectrophotometric and chromatographic analyses.

### 2.3. Preparation of Microgreens for Chromatographic Analysis

The microgreen species were cut and finely grounded using liquid nitrogen. These microgreen powders were extracted with 80% methanol (+0.1% HCl) (1:10 *w*/*v*) for 1 h with constant stirring on a mechanical shaker (Thys 2, MLW Labortechnik GmbH, Seelbach, Germany) [29]. After that, the samples were centrifuged at 4000× *g* for 10 min. Collected supernatants of broccoli (BC), amaranth (AM), and red beet (RB) microgreens were filtered through 0.22 µm filters and used for further characterization of their bioactive compounds by UHPLC Q-ToF MS. This extraction solvent was most commonly used for the extraction of bioactive compounds from plant materials [12,14,29,30,31,32], while Pintać et al. [30] showed that this solvent provided the highest yield of phenolic compounds. For additional characterization of highly sensitive glucosinolates (GLSs), the broccoli sample was extracted with boiled 70% methanol for 1 h on a thermoshaker (70 °C), because the activity of the enzyme myrosinase was inhibited in boiled methanol [10]. The obtained supernatant (BC1) was filtered and used for additional characterization of GLSs.

### 2.4. Preparation of Cold-Pressed Microgreen Juices for Chromatographic Analysis

Microgreen juices were passed through an SPE cartridge (CLEAN-UP^R^, C18 Extraction columns, Unendcapped-PKG50, UCT, Bristol, UK) before chromatographic analysis to remove sugars and other colloidal impurities. The SPE cartridge was conditioned by washing with 5 mL of acidified methanol (methanol containing 0.1% HCl) and milliQ water, respectively. After that, the samples were passed through the cartridge and washed with 5 mL of milliQ water. Adsorbed bioactive compounds were eluted with 1 mL of acidified methanol (methanol containing 0.1% HCl), filtered through 0.45 µm syringe filters, and analyzed by UHPLC Q-ToF MS.

### 2.5. UHPLC Q-ToF MS of Microgreens and Cold-Pressed Microgreen Juices

The phytochemical profiles of the microgreen extracts and prepared juices were analyzed using the Agilent 1290 Infinity ultra-high-performance liquid chromatography (UHPLC) system coupled with quadrupole time-of-flight mass spectrometry (6530C Q-ToF-MS) (Agilent Technologies, Inc., Santa Clara, CA, USA), according to the method described in detail previously by Kostić and Milinčić [33]. The QToF-MS system was equipped with a dual Agilent Jet Stream electrospray ionization (ESI) source that operated in both positive (ESI^+^) and negative (ESI^−^) ionization modes. The operating parameters for ESI were the same as previously reported by Kostić and Milinčić [33]. Agilent Mass Hunter software ver. 10.0 was used for instrument control, data acquisition, and analysis.

Individual glucosinolates, phenolic compounds, and betalains were identified based on their monoisotopic mass and MS fragmentation. In addition, data already published in the literature were also used for the identification of glucosinolates [9,34,35,36,37,38], phenolic compounds [10,31,39,40], and betalains [41,42,43,44]. Phenolic compounds were quantified by direct comparison with available standards. However, as specific phenolic derivatives were detected for which no specific standards exist, the amounts of the individual phenolic derivatives were quantified using available standards (sinapic acid for phenolic acid derivatives and apigenin for flavonoid derivatives), and expressed as mg/100 g fresh weight of microgreens (FW) or mg/100 mL juice. Appendix A shows a phenolic compounds used for quantification, together with their equation parameters and correlation coefficient (r^2^). The relative content of individual betalains in amaranth and red beet microgreens and juices (%) was calculated as the ratio of the area of each individual compound and the area of total compounds detected. The exact masses of the compounds were calculated using ChemDraw software (version 12.0, CambridgeSoft, Cambridge, MA, USA).

### 2.6. Proximate Compositions of Cold-Pressed Microgreen Juices

The pH of the microgreen juices was determined with a digital pH meter, while the total soluble solids (°Bx) were measured with a refractometer (ATC 0–32 Brix, Huixia Supply Co., Ltd., Fuzhou, China). The dry weight of the juices was determined gravimetrically by drying the samples at 105 °C to constant mass. Juice yield (%) was calculated as the ratio between the mass of obtained juice (*m*1) and the mass of fresh microgreens (*m*2) and calculated according to the following equation:(1)$$Yield\;of\;juice\;\left(\%\right)=\frac{m1}{m2}\times 100$$

### 2.7. Total Phenolics, Flavonoids, Betalains, and Chlorophyll Content in Microgreen Juices

The total phenolic content (TPC) and flavonoid content (TFC) of the microgreen juices were determined using the Folin–Ciocalteu colorimetric assay and the aluminum chloride assay [29]. Briefly, TPC was determined after reacting diluted juices (0.5 mL) with Folin–Ciocalteu reagent (2.5 mL) and 7.5% Na_2_CO_3_ (2.5 mL). For TFC, the juices (2 mL) were mixed and incubated with 5% NaNO_2_ (0.15 mL), 10% AlCl_3_ (0.15 mL), 1 M NaOH (1 mL), and milliQwater (1.2 mL), respectively. After incubation, the absorbance of the mixtures was measured at 760 nm (TPC) and 510 nm (TFC), using a UV-Vis spectrophotometer (model HALO DB-20S, Dynamica Scientific Ltd., Livingston, UK). Results for TPC and TFC were expressed as mg of gallic acid (mgGAE/100 mL) and quercetin (mgQE/100 mL) equivalents per 100 mL juice.

Total betacyanins and betaxanthins were determined according to the previously described method of Stintzing and Schieber [45]. Total betalains are the sum of total betacyanins and betaxanthins. The absorbance of the appropriately diluted microgreen juices was measured at 485 nm, 536 nm, and 650 nm. The results were expressed in mg per 100 mL juice and calculated according to the following equation:(2)$$Betaxanthins\left(Betacyanins\right)\;\left(\frac{\mathrm{mg}}{100\mathrm{mL}}\right)=\frac{A\times DF\times MW\times 100\;}{\varepsilon \times i}$$
where A = A_485_ − A_650_ (betaxanthins) and A = A_536_ − A_650_ (betacyanins). DF—dilution factor; MW—molecular weight (339 g/mol for betaxanthins and 550 g/mol for betacyanins); *ε*—molar extinction (48,000 for betaxanthins and 60,000 for betacyanins); *i*—the path length (cm).

Total chlorophyll *a* and chlorophyll *b* were calculated using Equations (2) and (3), respectively, as previously described by Ali and Popović [17]. Briefly, the microgreen juices were appropriately diluted with 80% acetone, while the absorbance of the prepared samples was measured at two wavelengths (645 nm and 663 nm).
(3)$$Chlorophyll\;a\;\left(\frac{\mathrm{mg}}{100\mathrm{mL}}\right)=\left(12.71\times A_{663}-2.59\times A_{645}\right)\times DF/10$$
(4)$$Chlorophyll\;b\;\left(\frac{\mathrm{mg}}{100\mathrm{mL}}\right)=\left(12.71\times A_{645}-2.59\times A_{663}\right)\times DF/10$$
where DF is the dilution factor, and A refers to the absorbances recorded at 645 nm and 663 nm. The results were expressed as mg per 100 mL of juice.

### 2.8. Antioxidant Properties of Cold-Pressed Microgreen Juices

The antioxidant properties of the microgreen juices were analyzed using three assays: DPPH radical scavenging activity assay (DPPH^•^ scavenging activity), ABTS radical cation scavenging activity assay (ABTS^•+^ scavenging activity), and ferric reducing antioxidant power assay (FRAP assay) [29,46]. Briefly, diluted juice (0.1 mL and 0.03 mL) was mixed with 1.9 mL of DPPH^•^ and 3 mL of ABTS^•+^ working solution, respectively. After incubation of the two reaction mixtures, the absorbance of the samples was measured at 515 nm (DPPH^•^) and 734 nm (ABTS^•+^). For the FRAP assay, the juice (0.1 mL) was mixed with 0.3 mL of milliQ water and 3 mL of FRAP reagents. The mixture was then incubated (37 °C, 40 min), and absorbance was recorded at 593 nm. Trolox was used as the standard for all antioxidant assays, and results were expressed as mg Trolox equivalent (TE) per 100 mL juice.

### 2.9. Sensory Properties of Cold-Pressed Microgreen Juices

Sensory evaluation (overall quality and consumer acceptance) of the microgreen juices was performed by trained evaluators/selected consumers at the Faculty of Agriculture, University of Belgrade. In addition, the sensory testing conditions were conducted in a controlled environment in accordance with ISO standards for sensory analysis, including ISO 8589:2007 [47] and ISO 11136:2014 [48]. The samples of microgreen juices were prepared as follows: (1) the selected microgreens were cut, weighed, washed, and cold-pressed using a super-slow juicer (Angel Juicer 8500, Angel Co., Ltd., Busan, Republic of Korea), (2) the obtained cold-pressed juices of microgreens (broccoli, amaranth, and red beet) were placed and served in a transparent glass to ensure good transparency of microgreen juices and coded with random three-digit numbers in an amount of 20 mL. The microgreen juices were tasted in the sensory room under the known conditions, such as controlled ventilation, with white light and separated from the sample preparation room and away from inappropriate odors and noise. Tap water and unsalted crackers were used as taste neutralizers after tasting each sample. The temperature of the juices was 8 °C, as for commercially available juices. The conditions of the sensory tests ensured that each evaluator/consumer could objectively award an appropriate score. All sensory tests with participants were conducted in accordance with the Code of Professional Ethics of the University of Belgrade [49]. Before sensory evaluation, all participants gave informed consent via the statement that they were aware that their responses were confidential, they agreed to participate in this study, their responses could be used, they could withdraw from the study at any time, and that there would be no release of participant data without their knowledge. The products tested were safe for consumption.

#### 2.9.1. Overall Quality Evaluation

The overall quality of the microgreen juices was determined using the quality assessment method, taking into account the following quality criteria: appearance, odor, texture, and flavor. The quality of the juices was assessed using category scales ranging from 0 (unsatisfactory quality) to 5 (excellent quality). This evaluation was performed using a 5-level quality scoring system described in [50].

As the sensory attributes have different effects on the overall quality of the juices, the following importance coefficients (CI) were used: 1 (appearance), 6 (odor), 5 (texture), and 8 (taste). To calculate the overall quality score for each evaluator, individual scores given to the selected sensory attributes were first multiplied by the corresponding CI, and then the sum of corrected score values was divided by the sum of CI.

The quality rating method was used to determine the overall quality of cold-pressed microgreen juices. Sensory evaluation using quality rating method was performed by trained evaluators who are employed at the Faculty of Agriculture and are well-versed in the process of producing cold-pressed beverages and vegetable/plant-based juices, and they also used a guide for sensory evaluation of quality microgreen juices. The 10 trained evaluators participated in this sensory testing (ISO 8586:2023) [51]. Before performing the quality rating method, all evaluators attended training sessions for three weeks for 2 h each and tasted the juices. The microgreen juices were presented to the evaluators monadically in random order.

#### 2.9.2. Consumer Acceptance Evaluation

Sensory acceptability of the microgreen juices by consumers was assessed using the 9-point hedonic scale (1–4: dislike; 5: neither like nor dislike; 6–9: like). The evaluators did not compare the samples with each other, but rated the sample according to individual sensory attributes and overall acceptability. The aim of the hedonic test is not the comparison of products or to range products by evaluator, but rather evaluation of the product in terms of overall acceptability and the similarity of the selected sensory attributes [52,53]. To ensure an absolutely independent evaluation, consumers did not assign scores to compare the results with each other or between samples. The first sensory characteristic that the evaluator rated was color, i.e., the appearance of the product: for example, “How much do you like the color of the product?”, and so on for each of the attributes. If the evaluator felt the need to describe any sensations related to the product or its attributes, they could write comments in the comment field. The numerical data obtained for the hedonic test were expressed as mean values in radar diagram. These data were statistically processed using a one-way ANOVA (Duncan’s post hoc test), which allowed a comparison of the juices with each other.

The prepared cold-pressed microgreen juices were served to consumers in clear plastic glasses, to help them perceive the color and appearance of the juices. The hedonic test was carried out on 74 consumers who regularly consume vegetable beverages. Criteria for the selection of consumers were as follows: (1) they must be consumers of these or similar products, and (2) they must have no health problems, in particular, no problems with oral perception or dysfunctional senses, or dental disfunction. The sensory panel included both men (57%) and women (43%) who consume different types of vegetable beverages, and the average age of evaluators was 29 years.

### 2.10. Statistical Analysis

Results for proximate compositions, spectrophotometric assays, chromatographic quantification, consumer acceptance evaluation (radar chart), and quality ranking were expressed as mean values ± standard deviation (*n* = 3). Significant differences between means were determined by one-way ANOVA, using Duncan’s post hoc test (IBM SPSS ver. 25 statistical software, SPSS Inc., Chicago, IL, USA). The correlation analysis was carried out by calculating Pearson’s correlation coefficient (*p* < 0.05).

## 3. Results and Discussion

### 3.1. UHPLC Q-ToF MS Profile of Bioactive Compounds of Broccoli, Amaranth, and Red Beet Microgreens

The major classes of bioactive compounds in broccoli, amaranth, and red beet microgreens were identified by UHPLC Q-ToF MS, taking into account the exact *m*/*z* masses of the molecular ions, typical MS fragments and available data in the literature [9,10,37,38,39,41,43,44]. Glucosinolates (GLSs) were detected only in broccoli microgreens, as expected. A total of seven GLSs compounds were detected in different broccoli microgreen extracts (Table 1).

The identification was performed in two different extracts of broccoli microgreens to obtain a better insight into the profiles of GLSs, as they are very unstable and rapidly hydrolyze to different degradation products [2,8]. Glucobrassicin and its various substituted hydroxy and methoxy derivatives were detected in both extracts, which is consistent with other studies that analyzed broccoli microgreens [11,54,55]. However, gluconapin and gluconasturtiin were detected in different extracts, depending on the extraction conditions and the extractant used. The absence of gluconapin in BC extract is probably due to its increased sensitivity to myrosinase activity. On the other hand, the presence of gluconasturtiin in the BC extract may be due to the different polarity of the extraction solvents used and the better tendency of acidified 80% methanol to extract this compound. Interestingly, glucoraphanin was not detected, which has been reported as the dominant GLS in broccoli microgreens in most studies [2,11,54]. Its absence in the extracts may be due to its lower stability, its rapid conversion into sulforaphane or its tendency to form conjugates with other compounds of the microgreens [56].

Phenolic compounds (PCs) were identified and quantified in the extracts of all microgreens studied. However, the phenolic compounds found in the analyzed microgreen species differed significantly (Table 2).

The highest total detected PC content is found in broccoli (1588.71 mg/100 g FW), followed by red beet (1331.01 mg/100 g FW), and the lowest is found in amaranth (646.38 mg/100 g FW) microgreen extracts. The high total PC content in amaranth and broccoli extracts is mainly due to various derivatives of phenolic acids (>95% of the total PC content), and in red beet microgreen extract, it is due to apigenin *C*-glycosides (>70% of the total PC content). Various hydroxybenzoic, dihydroxybenzoic, and vanillic acid glycosides were detected in all analyzed microgreen extracts, but their content varied and depended strongly on the microgreen species. For example, pentosyl hexoside glycosides of hydroxybenzoic, dihydrobenzoic, and vanillic acid were dominant in amaranth microgreens along with feruloyl isocitric (175.78 mg/100 g FW) and benzoyl malic acid. On the other hand, dipentosyl, and pentosyl glycosides of dihydroxybenzoic acid (252.35 and 59.79 mg/100 g FW) were the most abundant phenolic acid derivatives in red beet microgreens. Similarly to our results, Wojdyło and Nowicka [14] identified phenolic acids as the predominant class of phenolic compounds in amaranth microgreens. In addition, sinapic acid (526.06 mg/100 g FW) and its various derivatives (sinapic acid hexoside, sinapoyl malic acid, disinapoyl-dihexoside, and trisinapoyl-dihexoside) were the dominant compounds in broccoli microgreens. Identical sinapoyl derivatives were previously discovered and reported by Liu and Shi [10] in the analysis of differentially grown broccoli microgreens. In contrast to phenolic acids, the detected flavonoids can be characterized as specific markers for broccoli, amaranth, and red beet microgreens. Several apigenin *C*-glycosides, i.e., vitexin and cytisoside (3′-methyl vitexin) derivatives (10 compounds) were, to our knowledge, detected for the first time in red beet microgreens. Quantification confirmed that the total amount of cytisoside derivatives was significantly higher compared to vitexin derivatives, primarily contributed by compounds **32** (194.21 mg/100 g), **33** (184.04 mg/100 g), and **35** (170.18 mg/100 g) (Table 2). These compounds have a common MS base at 327 *m*/*z*, which is a typical fragment of the cytisoside molecule (*m*/*z* 447), obtained after cross-ring cleavage of the sugar unit. Compound 35 (*m*/*z* 651) was identified as 2″-hexosyl-6″-acetyl cytisoside. The key MS fragment (Y_0_^+^) for its identification was at 489 *m*/*z* ([M-acetyl residue-^0.2^X_8_ scission]^+^), followed by a fragment at 327 *m*/*z* ([Y_0_^+^-hexosyl residue (162Da)]^+^). Compounds **32** and **33** were recognized as 2″-hexosyl cytisoside (*m*/*z* 609) and 2″-pentosyl cytisoside (*m*/*z* 579), respectively, with MS secondary peak at 447 *m*/*z* obtained by the loss of hexosyl (−162 Da) or pentosyl (−132 Da) sugar units, respectively. In addition, both compounds yielded a fragment followed by another loss of 18 Da (H_2_O), indicating that the secondary sugar is linked to the primary sugar by an interglycosidic 1–2 bond. A typical MS^2^ fragment repeated in all cytisoside derivatives was found at 351 *m*/*z*, and its proposed structure is shown in Figure 2b. To date, only a few studies have confirmed the presence of several vitexin derivatives (compounds **27**, **28**, **29**, and **32** in Table 2) in mature red beet (*Beta vulgaris* L.) stalks and leaves [39,40,57]. Moreover, the cited studies have confirmed and characterized vitexin derivatives as potentially very useful compounds for human health, due to their antioxidant, anticancer and anti-inflammatory activities [40,57,58]. In contrast, apigenin *C*-glycoside was not found in broccoli and amaranth microgreens. Macromolecular kaempferol (compound **37**) and kaempferol-sinapoyl derivatives (compounds **38** and **39**) were identified only in broccoli microgreens and accounted for about 4% of the total quantified phenolics. These kaempferol derivatives were previously identified by Liu and Shi [10] and represent typical compounds found in *Brassica* microgreens and vegetables. Quercetin 3-*O*-(6″-rhamnosyl)-hexoside was found in all three analyzed microgreen species (amaranth, broccoli, and red beet). However, the highest content of this compound was found in amaranth microgreens (24.37 mg/100 g).

Based on a detailed analysis of betalains, the dominant presence of (iso)amarnthin (73.56%) and (iso)betanin (49.20%) was confirmed in amaranth and red beet microgreens, respectively (Table 3).

Both compounds produced a MS base peak at 389 *m*/*z* (betanidin aglycon), which resulted from the loss of glucosyl-glucuronyl residue ([M+H-176 Da-162 Da]^+^) for (iso)amaranthin and glucosyl unit ([M+H-162 Da]^+^) for (iso)betanin [41]. Other detected compounds are decorboxy derivatives of amaranthin (*m*/*z* 683) and betanin (*m*/*z* 507), with characteristic fragments at 345 *m*/*z* ([betanidin aglycone-44 Da (CO_2_)]^+^), 299 *m*/*z*, and 150 *m*/*z*. These betalain derivatives were previously identified and analyzed in detail in mature red beet root [43,44] or leaves/sprouts of various *Amaranthus* species [59]. However, to our knowledge, this is the first study to analyze betalain profiles in amaranth and red beet microgreens.

### 3.2. Proximate Composition of Microgreen Juices

Results for yield and general physicochemical parameters (moisture, dry weight, pH, and °Brix) of microgreen juices (BCJ, AMJ, and RBJ) are shown in Table 4. The yield of the microgreen juices varied from 53.4% (AMJ) to 70.2% (BCJ), which was a very high yield for the microgreen juices obtained by mechanical pressing. All microgreen juices had high moisture content (>98%) and low dry matter content (from 1.64 to 2.0%). The pH values of the microgreen juices ranged from 5.96 (BCJ) to 6.52 (AMJ), which is consistent with previously published results for the same microgreen species [24]. The values for total soluble solids (°Bx) are similar for all analyzed microgreen juices ranging from 1.8 to 2.0 °Bx. It can be concluded that these microgreen juices have a mild acidity and low sugar content, which places them among the low-calorie beverages.

### 3.3. UHPLC Q-ToF MS Profile of Microgreen Juices

In contrast to broccoli microgreens, GLSs were not detected in broccoli microgreen juice. During pressing of microgreens to juice, mechanical damage to the plant tissue occurs with the release of GLSs which are rapidly hydrolyzed to various isothiocyanates by the action of enzyme myrosinase [2]. Similarly to our results, Bello and Maldini [56] reported the complete absence of GLSs (glucoraphane) in broccoli sprout juices, obtained by mechanical pressing of raw and microwave-treated sprouts. However, various degradation products of GLSs (isothiocyanates) have also shown strong biological activity [8,15,22,23], and likely contribute to the antioxidant potential of BCJ along with phenolic compounds.

The prepared juices from amaranth, broccoli, and red beet microgreens (BCJ, AMJ, and RBJ) had high content of phenolic compounds, namely, 49.84 mg/100 mL, 362.37 mg/100 mL, and 342.02 mg/100 mL juice, respectively. As for microgreens (Table 2), phenolic acids and their derivatives were identified and quantified dominantly in BCJ and AMJ, while various apigenin *C*-glycosides were found mainly in RBJ. Based on the results presented in Table 2 and Table 5, the differences among phenolic acid derivatives detected in microgreens and their juices can be clearly observed. This is probably due to the different migration of individual phenolic compounds from the plant tissue into the juice or their transformation by enzyme action during juice production.

In addition, more phenolic acids (aglycones) and their derivatives with organic acids (quinic acid and malic acid) were identified in the juices of microgreens (Table 5), probably due to the increased contact between these molecules and their tendency to interact with each other. In amaranth microgreens and its juice, a high content of pentosyl hexoside glycosides of vanillic acid and dihydroxybenzoic acid was confirmed. However, some compounds that were dominantly detected in amaranth microgreens, such as feruloyl isocitric acid, benzoyl malic acid, and some glycosides of vanillic and hydroxybenzoic acids, were not detected in AMJ. It can be assumed that these compounds were degraded or transformed during juice production. This is partially supported by the fact that the high content of hydroxybenzoic (14.97 mg/100 mL) and benzoic (3.68 mg/100 mL) acid was detected only in amaranth juice (AMJ). A wide variety of phenolic acids derivatives were identified in BCJ. Commonly present sinapic acid derivatives (sinapic acid, sinapoyl malic acid, and sinapic acid hexoside) were detected in high amounts in both broccoli microgreens and juice (Table 2 and Table 5). However, the characteristic di- and trisinapoyl glycosides previously detected in broccoli microgreens [10], were not found in BCJ, so it can be assumed that these macromolecules were retained in the residues generated during the pressing of broccoli juice. In addition, high levels of hydroxybenzoic acid and dihydroxybenzoic acid along with benzoic acid derivatives were also detected in BCJ, while their glycoside forms were present only in trace amounts. It is worth mentioning that kaempferol-3-*O*-sinapoyl-dihexoside-7-*O*-hexoside was the only flavonoid detected in BCJ, with a content of 4.77 mg/100 mL. In contrast to AMJ and BCJ, low levels of phenolic acid derivatives (about 5.5% of the total amount of PCs) were detected in red beet microgreen juice (Table 5). Among the individual phenolic acids, dihydroxybenzoic acid pentosyl hexoside (similar to that in BR) and hydroxybenzoic acid were the dominant compounds in this juice, with contents of 7.33 mg/100 mL and 9.81 mg/100 mL, respectively. Other phenolic acid derivatives were found in small or trace amounts. Interestingly, dihydroxybenzoic acid dipentoside, the predominant compound in red beet microgreens, was not detected in the juice. This compound was probably retained in the solid waste of the microgreens or converted to other phenolic acid derivatives during juice production. However, the juice of red beet microgreens is a good source of several apigenin *C*-glycosides (Table 5), which appear to be highly soluble and readily transferred from the microgreens to the juice. A total of 13 vitexin and cytisoside derivatives were identified, accounting for 94.4% of the total amount of PCs in RBJ. Cytisoside derivatives dominated, especially 2″-hexosyl-6″-acetyl cytisioside (67.44 mg/100 mL), followed by hexosyl cytisoside (48.94 mg/100 mL) and pentosyl cytisoside (46.00 mg/100 mL), just as in red beet microgreens. In addition, 2″-hexuronyl-6″-acetyl cytisoside (compound **36**, Table 5) was also confirmed at a high level (18.49 mg/mL), but only in RBJ. The proposed fragmentation pathway for this compound is shown in Figure 2b. Among the identified vitexin derivatives, special attention should be paid to 2″-hexosyl-6″-malonyl vitexin, 2″-hexosyl vitexin and 2″-pentosyl vitexin, whose individual contents exceeded 25 mg/100 mL. Several studies have already identified these vitexin derivatives in red beet leaves and pointed out their numerous health benefits [39,40,57,58]. Other apigenin C-glycosides were present in significantly lower amounts. Chromatograms extracted on the accurate masses of dominant cytisoside and vitexin derivatives are shown in Figure 2a. Finally, to the best of our knowledge, this study marked the first time that detailed profiles of the phenolic compounds of microgreen juices were obtained, making comparisons with other studies difficult.

In addition to the phenolic compounds, several characteristic betalains were identified in RBJ and AMJ (Table 6).

The good water solubility of betalains contributes to their effective and rapid leaching (migration) from the tissues to juice. Amaranthin and isoamaranthine were detected mainly in AMJ. In contrast, the most abundant betalains in red beet juice were betaine and isobetanin. Similarly to our results, Sawicki and Martinez-Villaluenga [43] reported a high total content of betalains in fresh red beet juice, with betanin dominating. In both microgreen juices (AMJ and RBJ), betalamic acid was identified as the third dominant compound. Betalamic acid is a precursor in the synthesis of betalains, which explains its presence in these microgreen juices. On the other hand, betanin can form complexes with feruloyl residues, which explains the presence of 6′-*O*-feruloyl-betanin in RBJ. This compound was previously found in dried red beet [44] and other red beet products [43]. The relative content of decarboxy derivatives of amaranthine, betanin, and neobetanin was low in both juices (<3% of the total betalains). This means that the betalains remained as carriers of the red color in these cold-pressed microgreen juices. A higher content of decarboxy derivatives is characteristic of thermally treated juices, resulting in a reduction in betalains, an increased content of decarboxy derivatives of betalains, and the appearance of a brown color [60]. Betaxanthins (two compounds) were found only in red beet juice (RBJ), but their relative content was low (<4% of the total betalains and betaxanthins).

### 3.4. Total Phenolic, Flavonoid, Betalain, and Chlorophyll Content in Cold-Pressed Microgreen Juices

The spectrophotometrically determined total phenolic and flavonoid content of the microgreen juices is shown in Figure 3a.

The highest TPC value was obtained for broccoli juice (92.92 mg GAE/100 mL), followed by red beet (71.26 mg GAE/100 mL) and amaranth (50.86 mg GAE/100 mL) microgreen juices, which showed the same trend as the results of chromatographic analysis. The obtained results are in the range of TPC values previously reported by other authors for differently processed *Alternanthera sessilis* [16] and wheatgrass [18] microgreen juices. The TFC values for amaranth, red beet and broccoli microgreen juices were 45.94, 46.35 and 61.56 mg QE/100 mL, respectively. As shown in Figure 3a, the TFC values for RBJ and AMJ were not significantly different. These TFC results do not agree with the results of chromatographic analysis, which showed the dominant presence of flavonoids in RBJ, a low level in BCJ and traces in AMJ. However, the interpretation of these results should take into account the limitation of the spectrophotometric TFC method. Finally, the obtained variations in TFC values can be explained by the following facts: (1) the most abundant apigenin derivatives in RBJ do not contribute to the absorbance at 510 nm, and (2) phenolic acids show considerable absorbance at the same wavelength [61].

Pigments are responsible for the appealing and attractive color of microgreens. The intense color of amaranth and red beet juices comes from betalain pigments, which have various health benefits. Amaranth microgreen juice had significantly higher levels of total betalains (36.73 mg/100 mL), betacyanins (27.39 mg/100 mL), and betaxanthins (9.34 mg/100 mL), compared to red beet microgreen juice (Figure 3b). However, the obtained results suggest that the prepared cold-pressed microgreen juices are a good source of betalains. Moreover, betacyanins were predominantly detected in both juices (RBJ and AMJ), almost three times more than betaxanthins. Betalains were not detected in broccoli juice. These results are in agreement with the chromatographic profiles (Table 6). On the other hand, the green color of microgreens comes from chlorophylls, and the green hue of plant samples is often directly proportional to the amount of these pigments. The highest levels of total chlorophyll (20.19 mg/100 mL), chlorophyll *a* (14.60 mg/100 mL), and chlorophyll *b* (5.60 mg/100 mL) were found in red beet microgreen juice, while the lowest levels were found in broccoli microgreen juice (Figure 3c). Chlorophylls are unstable pigments that are readily converted to pheophytin (dark green/brown color) at an acidic pH, and this conversion may occur primarily in cold-pressed broccoli microgreen juices.

### 3.5. Antioxidant Properties of the Cold-Pressed Microgreen Juices

The results of the antioxidant assays for the microgreen juices are shown in Figure 3d. All microgreen juices prepared had good ABTS^•+^ scavenging activity. The highest activity against ABTS^•+^ was shown by broccoli microgreen juice (143.38 mg TE/100 mL), to which phenolic compounds and isothiocyanates probably contribute. On the other hand, the obtained values of DPPH^•^ scavenging activity for all microgreen juices were significantly lower (about 7-fold) compared to the ABTS^•+^ values. The prepared microgreen juices mostly contain hydrophilic biocompounds, which apparently have a greater affinity to interact with more polar ABTS radical cations than with the hydrophobic DPPH radicals. Similar observations and results were reported by Skoczylas and Korus [18], who analyzed the ABTS^•+^ and DPPH^•^ scavenging activity of fresh and frozen wheatgrass juice. Moreover, all microgreen juices had a high tendency to reduce the [Fe^3+^-(TPTZ)_2_]^3+^ complex, indicating their good reduction capacity. The highest FRAP value was obtained for BCJ, followed by RBJ and AMJ. Correlation analysis revealed that FRAP of the microgreen juices had a strong positive correlation with TPC (*r* = 0.98), TFC (*r* = 0.96) and total phenolic acids (*r* = 0.93).

### 3.6. Sensory Properties of Cold-Pressed Microgreen Juices

The results for the sensory quality evaluation of microgreen juices produced are shown in Figure 4a.

The mean quality scores for odor, texture, and overall quality did not significantly differ for all analyzed microgreen juices. Amaranth microgreen juice had the highest score for texture (4.7) due to its completely clear appearance and absence of colloidal cloudiness. All analyzed juices had a typical odor, characteristic of the respective microgreens and plant species. On the other hand, the mean value for the appearance of broccoli juice was significantly lower (*p* < 0.05) compared to the juices from red beet and amaranth microgreens. According to the comments of evaluators, the low score for the appearance of broccoli juice was due to high colloidal turbidity and rapid precipitation of particles. Amaranth and red beet microgreen juices received excellent scores for appearance due to their light purple/red color, which is characteristic of these microgreen species. Evaluators provided critical comments and low scores for taste, ranging from 4.0 (BCJ) to 2.9 (RBJ). The main deficiencies of amaranth and red beet microgreen juices were a slight astringency and an earthy and stale flavor, while broccoli microgreen juice had a flavor typical of plants from the *Brassicaceae* family, with herbaceous, grassy, and sulfurous notes. The overall quality of the microgreen juices was rated 3.5 to 4.5 points, indicating good acceptability and sensory quality of these juices.

The results for overall acceptability of the microgreen juices ranged from 5.0 (neither like nor dislike) to 6.4 (slightly like) (Figure 4b). The broccoli microgreen juice received the lowest rating for overall acceptability. The evaluators pointed out the bitter and astringent taste of broccoli juice, which was probably due to the phenolic acids and GLSs metabolites (isothiocyanates) that were predominantly present. In addition, the score of overall acceptability showed a strong positive correlation with the results for taste (*r* = 0.771) and odor (*r* = 0.611). This indicates that these sensory attributes had the greatest influence on the overall acceptance of the microgreen juices by consumers. The high overall acceptance and good appearance of the amaranth microgreen juice was probably mainly influenced by its intense color. Similar observations were reported by other authors [24,25], who studied the sensory properties of broccoli, amaranth and red beet microgreens.

## 4. Conclusions

In summary, the prepared juices from amaranth, broccoli, and red beet microgreens had a high content of phenolic compounds, but this was significantly lower than in the corresponding extracts of microgreens. In addition, the composition of the bioactive compounds in the juices differed significantly from their profile in the starting materials. Various glucosinolates have been detected in broccoli microgreens. However, they were not found in broccoli microgreen juice, as they are degraded to isothiocyanates. Further, more phenolic acid aglycones and their derivatives with organic acids (quinic acid and malic acid) were detected in the juices compared to extracts of microgreens. This is probably due to the differential migration of individual phenolic compounds from the plant tissue into the juice or their transformation due to enzyme action and increased contact between the molecules during the production process. Phenolic acids, especially sinapic acid and its derivatives (sinapoyl malic acid and sinapic acid hexoside), were predominantly found in broccoli microgreen juice, while pentosyl hexoside glycosides of vanillic acid and dihydroxybenzoic acid were most frequently detected in amaranth microgreen juices. Low levels of phenolic acid derivatives were found in red beet microgreen juice. However, red beet microgreens and juices were a good source of apigenin *C*-glycosides (vitexin and cytosioside derivatives), which apparently readily transfer from the microgreens to the juice. Betalains were detected in both red beet and amaranth microgreens and juices, with betanin and decarboxy betanin dominating in red beet microgreens/juice, and amaranthin dominating in amaranth microgreens/juice. Interestingly, betalamic acid and some betaxanthins were detected, although at low levels, only in red beet and amaranth juices.

All microgreen juices exhibited good ABTS^•+^ scavenging activity and ferric reducing antioxidant power (FRAP). Considering the previous characterizations, it can be concluded that phenolic acid derivatives/isothiocyanate, phenolic acid derivatives/amaranthin, and apigenin-C-glycoside/betanin have the greatest influence on the functional properties of broccoli, amaranth, and red microgreen juices, respectively. In addition, these microgreen juices showed good overall quality and overall acceptability. Cold-pressing resulted in a high yield of microgreen juices characterized by mild acidity and low total soluble solids, which means they can be recommended as novel functional and low-calorie beverages. However, the specific flavor and astringency were the main drawbacks of the prepared microgreen juices. Therefore, it would be desirable to produce and characterize microgreen/fruit-based juices in future studies to mask the flavor and improve their sensory properties.

## Figures and Tables

**Figure 1 foods-13-00757-f001:**
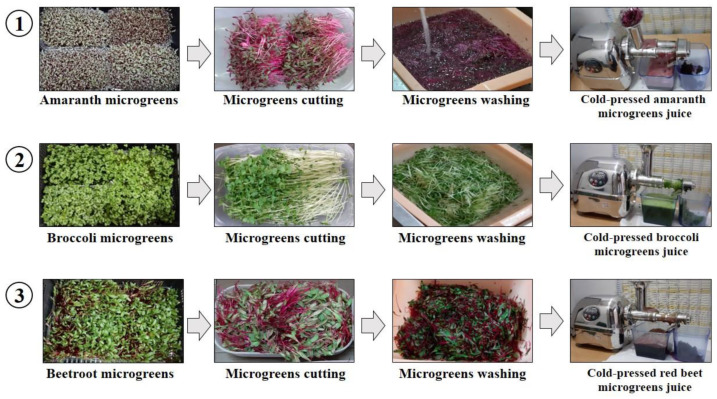
Schematic representation of cold-pressed juices preparation from amaranth, broccoli, and red beet microgreens.

**Figure 2 foods-13-00757-f002:**
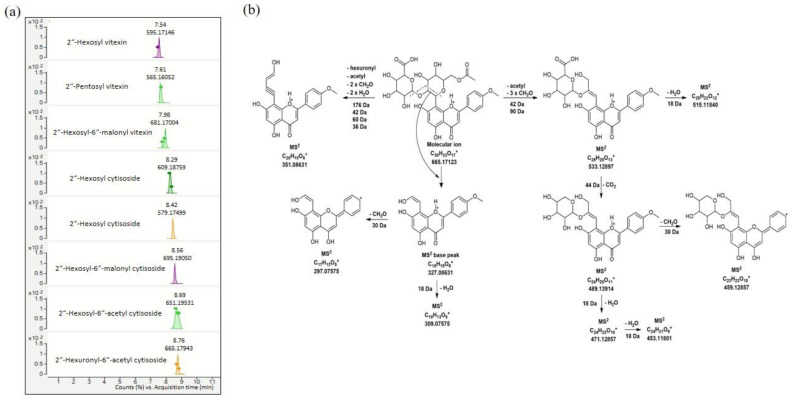
Chromatograms of the dominant cytisoside and vitexin derivatives in red beet cold-pressed juice, with retention time and exact mass (**a**); proposed fragmentation pathway of 2″-Hexuronyl-6″-acetyl cytisoside (**b**).

**Figure 3 foods-13-00757-f003:**
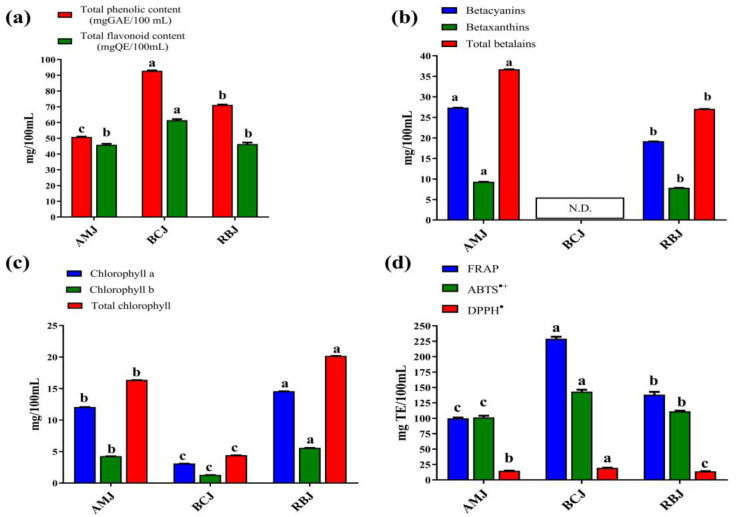
Total phenolic and flavonoid content (**a**), total betalain and betaxanthin content (**b**), total chlorophylls content (**c**), and antioxidant properties (**d**) of microgreen juices. Different lowercase letters denote significant differences between the microgreen juices, according to Duncan’s test (*p* < 0.05). Abbreviations: AMJ—cold-pressed amaranth microgreen juice; BCJ—cold-pressed broccoli microgreen juice; RBJ—cold-pressed red beet microgreen juice.

**Figure 4 foods-13-00757-f004:**
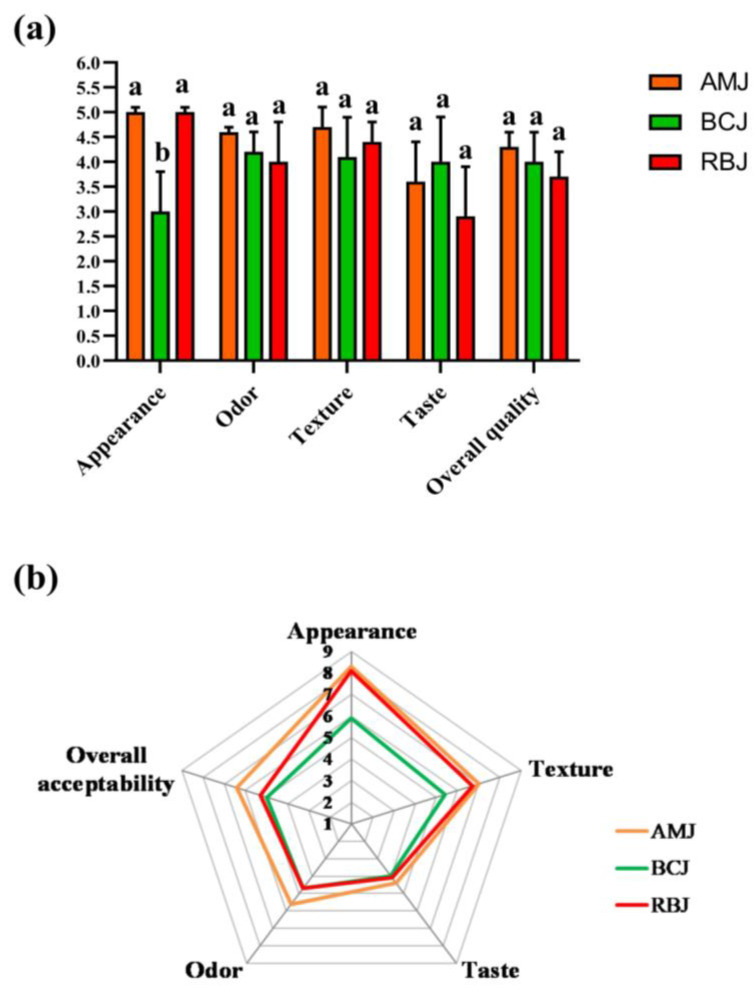
(**a**) Sensory quality scores; (**b**) sensory radar chart for the likeability testing of amaranth, red beet, and broccoli microgreen juices. Different lowercase letters denote a significant difference between the microgreen juices, separately evaluated for each sensory attribute, according to Duncan’s test (*p* < 0.05). Abbreviations: AMJ—cold-pressed amaranth microgreen juice; BCJ—cold-pressed broccoli microgreen juice; RBJ—cold-pressed red beet microgreen juice.

**Table 1 foods-13-00757-t001:** Characterization of glucosinolates in broccoli microgreens by UHPLC-QToF-MS. Target compounds, expected retention time (RT), molecular formula, calculated mass, exact mass, and MS fragments are presented.

RT	Formula	Calculated Mass	mDa	Compound Name	*m*/*z* Exact Mass	Major MS Fragments (Base Peak)	BC	BC1	Ref
1.72	C_11_H_18_NO_9_S_2_^−^	372.0423	−5.00	Gluconapin	372.0473	130(100), **195**, **259**, **275**, 241, 291, 139	**-**	**+**	[9,37,38]
6.71	C_15_H_20_NO_9_S_2_^−^	422.0579	−2.60	Gluconasturtiin	422.0605	205(100), 247, 164, **259**, **275**, 180, 226, 244, 342	**+**	**-**	
6.14	C_16_H_19_N_2_O_9_S_2_^−^	447.0532	−4.15	Glucobrassicin	447.0574	130(100), **259**, **205**, 447, **275**, 165, **195**	**+**	**+**	
1.77	C_16_H_19_N_2_O_10_S_2_^−^	463.0481	−4.35	4-Hydroxy- glucobrassicin	463.0525	169(100), 160, 221, **259**, **275**, **195**, 463, **205**, 285, 383, 267, 241, 186, 176	**+**	**+**	
2.56	C_16_H_19_N_2_O_10_S_2_^−^	463.0481	−4.35	5-Hydroxy- glucobrassicin	463.0525	169(100), 160, 221, **259**, 267, **275**, **195**, **205**, 285, 463	**+**	**+**	
7.13	C_17_H_21_N_2_O_10_S_2_^−^	477.0638	−4.54	Neo- glucobrassicin	477.0683	477(100), **259**, **275**, 284, 235, **195**, 241, 145	**+**	**+**	
7.88	C_17_H_21_N_2_O_10_S_2_^−^	477.0638	−4.54	4-Methoxy- glucobrassicin	477.0683	167(100), **259**, **205**, 241, **275**, 282, 285, **195**, 315, 447	**+**	**+**	

Abbreviations: BC—broccoli microgreens extracted with acidified 80% methanol at room temperature; BC1—broccoli microgreens extracted with 70% boiling methanol, at thermoshaker. “+”—detected glucosinolates.

**Table 2 foods-13-00757-t002:** Characterization and quantification (mg/100 g) of phenolic compounds detected in amaranth, red beet, and broccoli microgreens by UHPLC-QToF-MS. Target compounds, expected retention time (RT), molecular formula, calculated mass, exact mass, and MS fragments are presented.

No	Compounds Name	RT	Formula	Calculated Mass	*m*/*z* Exact Mass	mDa	MS Fragments (% of Base Peaks)	Samples (mg/100 g)		
								AM	BC	RB
**Phenolic acid and derivatives**										
1	Dihydroxy- benzoic acid hexoside isomer I ^b^	3.83	C_13_H_15_O_9_^−^	315.07216	315.07504	−2.88	**108.0228 (100)**, 109.0298 (41), 110.0330 (4), 152.0132 (58), 153.0199 (19), 154.0198 (2)	**10.08 ± 0.08 ^B^**	**104.14 ± 1.14 ^A^**	**/**
2	Vanillic acid hexoside isomer I ^b^	4.45	C_14_H_17_O_9_^−^	329.08781	329.08943	−1.63	**108.021 (100)**, 109.0263 (8), 113.0218 (4), 123.0449 (35), 124.0473 (3), 125.0240 (8), 152.0109 (76), 153.0136 (10), 167.0364 (34), 169.019 (3)	**4.23 ± 0.03 ^B^**	**11.25 ± 0.06 ^A^**	**/**
3	Hydroxy- benzoic acid dihexoside ^b^	5.18	C_24_H_19_O_10_^−^	467.09837	467.10054	−2.17	**137.0246 (100)**, 138.0278 (9), 299.0771 (2), 431.1188 (5)	**25.08 ± 0.10**	**/**	**/**
4	Vanillic acid hexoside isomer II ^b^	5.18	C_14_H_17_O_9_^−^	329.08781	329.08927	−1.46	**108.0229 (100)**, 109.0265 (8), 122.0367 (3), 123.0464 (39), 124.0504 (4), 152.0129 (61), 153.0172 (4), 167.0369 (32), 168.0409 (4)	**7.34 ± 0.04 ^B^**	**38.53 ± 0.54 ^A^**	**4.41 ± 0.12 ^C^**
5	Hydroxy- benzoic acid pentosyl hexoside isomer I ^b^	5.52	C_18_H_23_O_12_^−^	431.11950	431.12157	−2.07	**137.0246 (100)**, 138.0287 (8)	**95.15 ± 0.37 ^A^**	**/**	**2.58 ± 0.08 ^B^**
6	Carboxy- vanillic acid ^b^	5.59	C_9_H_7_O_6_^−^	211.02430	211.02751	−3.21	108.0235 (39), 109.0275 (2), 121.0287 (2), **122.0386 (100)**, 123.0453 (35), 124.0465 (3)	**/**	**28.99 ± 0.22**	**/**
7	Dihydroxy- benzoic acid hexoside isomer II ^b^	5.65	C_13_H_15_O_9_^−^	315.07216	315.07845	−6.29	108.0222 (8), **109.0305 (100)**, 110.0335 (14), 152.0114 (12), 153.0208 (61)	**10.33 ± 0.12 ^B^**	**19.23 ± 0.20 ^A^**	**3.83 ± 0.03 ^C^**
8	Vanillic acid pentosyl hexoside ^b^	5.79	C_19_H_25_O_13_^−^	461.13007	461.13684	−6.77	108.0226 (5), 123.0448 (8), 152.0113 (18), 153.0166 (2), **167.0356 (100)**, 168.0383 (10)	**123.40 ± 0.83**	**/**	**/**
9	Sinapoyl syringic acid ^b^	5.92	C_20_H_19_O_9_^−^	403.10290	403.10103	1.87	138.0306 (51), 153.0543 (31), 154.0590 (12), 161.0362 (27), 182.0204 (34), 189.0316 (89), 190.0316 (8), **197.0445 (100)**, 198.0494 (13), 203.0441 (7), 204.0558 (15)	**/**	**7.93 ± 0.05**	**/**
10	Dihydroxy- benzoic acid pentosyl hexoside ^b^	6.05	C_18_H_23_O_13_^−^	447.11442	447.11850	−4.08	101.0249 (4), 108.022 (14), 109.0298 (15), 151.0397 (3), **152.0118 (100)**, 153.0169 (13), 154.0189 (1), 161.0464 (3), 315.0738 (2), 447.1157 (37)	**65.56 ± 0.98 ^A^**	**/**	**4.55 ± 0.15 ^B^**
11	Dihydroxy- benzoic acid pentoside ^b^	6.05	C_12_H_13_O_8_^−^	285.06159	285.06344	−1.85	**108.0221 (100)**, 109.0282 (19), 152.0118 (43), 153.0171 (9)	**2.65 ± 0.06 ^B^**	**/**	**59.79 ± 1.89 ^A^**
12	Syringic acid hexoside ^b^	6.32	C_15_H_19_O_10_^−^	359.09837	359.10153	−3.15	101.0242 (59), 113.0246 (64), 121.0289 (38), 137.0261 (45), 138.0326 (71), 152.0488 (52), 153.0559 (56), 166.0279 (18), 181.0140 (41), 182.0234 (56), 196.0406 (22), **197.0449 (87)**, 211.0609 (20), 239.0567 (34), **359.1005 (100)**	**/**	**54.42 ± 0.98 ^A^**	**17.16 ± 0.26 ^B^**
13	Coumaric acid pentoside ^b^	6.32	C_14_H_15_O_7_^−^	295.08180	295.08967	−7.87	108.0206 (22), 127.1141 (13), 149.0220 (24), 151.1095 (19), 152.0415 (13), **163.0405 (100)**	**/**	**15.80 ± 0.24**	**/**
14	Hydroxy- benzoic acid hexoside ^b^	6.45	C_13_H_15_O_8_^−^	299.07724	299.08261	−5.37	108.0823 (8), 121.024 (11), 122.0373 (61), 123.0458 (51), **137.0252 (100)**, 138.027 (10)	**9.44 ± 0.04 ^C^**	**10.02 ± 0.13 ^B^**	**25.76 ± 0.54 ^A^**
15	Dihydroxy- benzoic acid dipentoside ^b^	6.66	C_17_H_21_O_12_^−^	417.10385	417.10765	−3.80	108.022 (18), 109.0298 (23), 110.0326 (1), 151.0402 (4), **152.0121 (100)**, 153.0168 (13), 285.0628 (2), 417.1050 (25)	**/**	**/**	**252.35 ± 2.32**
16	Caffeic acid hexoside ^b^	6.66	C_15_H_17_O_9_^−^	341.08730	341.09286	−5.56	134.0356 (5), **135.0466 (100)**, 136.0467 (9), 137.0575 (11), 145.0860 (5), 161.0255 (6), 164.0495 (6), 178.0266 (3), 179.0352 (61)	**/**	**16.47 ± 0.08**	**/**
17	Feruloyl quinic acid ^b^	6.86	C_17_H_19_O_9_^−^	367.10290	367.10878	−5.88	111.0471 (4), 117.0343 (15), 120.9974 (9), **134.0387 (100)**, 135.0427 (14), 146.0620 (11), 149.0617 (8), 155.0341 (5), 173.0472 (5), 190.0531 (6), 191.0589 (6), **193.0521 (49)**, 194.0527 (6)	**/**	**21.67 ± 0.24**	**/**
18	Sinapic acid dihexoside ^b^	6.80	C_23_H_31_O_15_^−^	547.16630	547.17128	−4.98	101.0242 (22), 113.0239 (10), 119.0346 (12), 149.0263 (9), 164.0477 (26), 179.0676 (13), 190.0262 (18), 205.0523 (80), 206.0563 (27), 221.0789 (40), **223.0616 (69)**, **247.0621 (100)**	**/**	**35.71 ± 0.48**	**/**
19	Ferulic acid hexoside ^b^	7.07	C_16_H_19_O_9_^−^	355.10346	355.10802	−4.56	**111.009 (100)**, 112.0128 (7), 132.0233 (6), 134.0375 (85), 135.0408 (8), 149.0616 (20), 154.9994 (12), 160.0170 (73), 161.0204 (8), 175.0402 (59), 176.0450 (8), 178.0277 (44), 179.0308 (5), **193.0514 (28)**, 194.0534 (4)	**39.70 ± 0.14 ^A^**	**7.80 ± 0.13 ^B^**	**/**
20	Sinapic acid hexoside ^b,^***	7.14	C_17_H_21_O_10_^−^	385.11402	385.11743	−3.41	101.0262(2), 113.0255(2), 119.0222(2), 147.0115(2), 149.0268(4), 164.0499 (10), 175.0056 (12), 190.0296 (100), 191.0334 (13), 192.03511(2), **205.05315 (77)**, 206.0571 (13), 207.0594 (2), 223.0633 (3)	**/**	**214.46 ± 2.22**	**/**
21	Feruloyl isocitric acid ^b^	8.14	C_16_H_15_O_10_^−^	367.06707	367.07166	−4.59	**111.0092 (100)**, 112.0128 (7), 129.0199 (3), 134.0379 (2), 154.9992 (12), 173.0099 (3)	**175.78 ± 1.10 ^A^**	**/**	**6.59 ± 0.09 ^B^**
22	Sinapoyl malic acid ^b,^***	8.35	C_15_H_15_O_9_^−^	339.07216	339.07690	−4.74	115.0054 (34), 116.0085 (2), 121.0311 (8), 132.0237 (2), **133.0161 (29)**, 134.0199 (2), 147.0469 (7), **149.0263 (100)**, 150.0297 (10), 164.0499 (79), 165.0532 (9), 193.0168 (2), 208.0406 (3), **223.064 (6)**	**/**	**159.12 ± 1.90**	**/**
23	Sinapic acid ^a^	8.35	C_11_H_11_O_5_^−^	223.06120	223.06408	−2.89	104.0281 (4), 117.0361 (3), **121.0309 (100)**, 122.0342 (10), 132.0250 (1), 135.0460 (3), 149.0257 (62), 150.0288 (6), 163.0413 (2), 165.0227 (4), 193.0166 (9)	**/**	**526.06 ± 2.29**	**/**
24	Benzoyl malic acid ^b^	8.48	C_11_H_9_O_6_^−^	237.03990	237.04732	−7.42	103.4469 (3), 115.0112 (3), **121.0295 (100)**, 122.0361 (8)	**53.27 ± 1.27 ^A^**	**4.45 ± 0.12 ^B^**	**/**
25	Disinapoyl-dihexoside ^b,^***	8.75	C_34_H_41_O_19_^−^	753.22420	753.23231	−8.11	119.0359 (3), 164.0496 (4), 179.0659 (3), 190.0294 (7), **205.0529 (100)**, 206.0565 (14), 208.0398 (3), 223.0637 (66), 224.0672 (8), 247.0642 (9), 265.0760 (4), 289.0751 (3), 529.1625 (42), 530.1661 (14), 531.1661 (3)	**/**	**122.51 ± 1.95**	**/**
26	Trisanapoyl-dihexoside ^b,^***	9.29	C_45_H_51_O_23_^−^	959.28210	959.28907	−6.97	**205.0525 (75)**, 206.0564 (7), 223.0637 (31), 247.0641 (14), 265.0763 (7), 289.0759 (7), 511.1509 (28), 512.1537 (8), 529.1607 (30), 530.1613 (9), **735.2217 (100)**, 736.2243 (46), 737.2255 (13), 959.2905 (14)	**/**	**130.53 ± 1.96**	**/**
**∑**								**622.02**	**1529.08**	**377.03**
**Apigenin C-glycosides ******										
27	2″-Hexosyl vitexin ^c^	7.82	C_27_H_31_O_15_^+^	595.16630	595.17351	−7.21	271.0608 (32), 283.0597 (17), 295.06144 (13), **313.0709 (100)**, 314.0746 (25), 337.0715 (20), 367.0819 (10), 379.0809 (9), 397.0928 (28), 398.0949 (9), 415.1031 (48), 416.1061 (14), 433.1138 (87), 434.1180 (26), 435.1191 (7)	**/**	**/**	**65.09 ± 0.31**
28	2″-Pentosyl vitexin ^c^	7.95	C_26_H_29_O_14_^+^	565.15570	565.16177	−6.07	283.0599 (16), 295.0603 (6), **313.0714 (100)**, 314.0753 (24), 337.0711 (16), 343.0821 (7), 367.0818 (14), 379.0818 (11), 397.0923 (42), 398.097 (12), 415.1031 (68), 416.1074 (19), 433.1138 (90), 434.1172 (28), 565.1550 (14)	**/**	**/**	**87.04 ± 1.02**
29	2″-Hexosyl-6″-malonyl vitexin ^c^	8.22	C_30_H_33_O_18_^+^	681.16670	681.17274	−6.04	271.0606 (19), 283.0604 (7), 295.0610 (21), **313.0712 (100)**, 314.0739 (24), 337.0706 (13), 345.1099 (7), 439.1031 (9), 457.1122 (8), 475.1238 (13), 483.0918 (7), 501.1043 (22), 502.1063 (8), 519.1149 (60), 520.1182 (23)	**/**	**/**	**67.35 ± 2.12**
30	2″-Hexosyl-6″-acetyl vitexin ^c,^*	8.27	C_29_H_31_O_16_^−^	635.16120	635.16454	−3.34	101.0257 (5), 175.0376 (6), 193.0516 (8), **293.0464 (100)**, **311.0613 (15)**, 337.0938 (7), 413.0880(21), 431.0900(5), **455.0981 (88)**, 473.1113 (12), 575.1456 (9)	**/**	**/**	**6.97 ± 0.06**
31	2″-Pentosyl-6″-malonyl vitexin ^c^	8.36	C_29_H_31_O_17_^+^	651.15610	651.15914	−3.04	283.0600 (9), 295.0611 (19), **313.0713 (100)**, 314.0744 (23), 337.0707 (13), 379.0820 (8), 439.1029 (11), 445.1139 (8), 457.1149 (10), 475.1246 (7), 483.0936 (7), 501.1035 (20), **519.1146 (41)**, 520.1178 (15), 651.1564 (11)	**/**	**/**	**76.12 ± 0.79**
32	2″-Hexosyl cytisoside ^c^	8.49	C_28_H_33_O_15_^+^	609.18190	609.18544	−3.54	285.0764 (34), 297.0764 (15), 309.0762 (14), **327.0874 (100)**, 351.0868 (18), 381.0977 (9), 393.0975 (8), 411.1082 (28), 429.1191 (47), **447.1296 (88)**	**/**	**/**	**170.18 ± 1.63**
33	2”-Pentosyl cytisoside ^c^	8.69	C_27_H_31_O_14_^+^	579.17140	579.17423	−2.83	297.0767 (15), **327.0872 (100)**, 328.0908 (25), 351.0863 (15), 357.0970 (9), 381.0975 (14), 393.0975 (10), 411.1088 (41), 412.1115 (13), 429.1193 (70), 430.1223 (23), **447.1296 (94)**, 448.1334 (31), 579.1716 (18), 580.1764 (7)	**/**	**/**	**184.04 ± 1.83**
34	2”-Hexosyl-6”-malonyl cytisoside ^c,^**	8.83	C_31_H_35_O_18_^+^	695.18230	695.18826	−5.96	285.0763 (24), 297.0759 (9), 309.0767 (20), **327.0869 (100)**, 328.0904 (25), 351.0871 (12), 393.0976 (8), 453.1185 (10), 471.1286 (10), 489.1415 (15), 497.1094 (7), 515.1202 (24), 516.1235 (9), **533.1299 (76)**, 534.1330 (28)	**/**	**/**	**64.77 ± 1.35**
35	2″-Hexosyl-6″-acetyl cytisoside ^c^	8.96	C_30_H_35_O_16_^+^	651.19250	651.19507	−2.57	297.0764 (6), **327.0871 (7)**, 369.0976 (22), 370.1010 (7), 381.0966 (4), 393.0958 (4), 411.1094 (12), 423.1086 (4), 429.1180 (10), 430.1229 (4), 471.1295 (14), 472.1314 (5), **489.1406 (100)**, 490.1438 (36), 491.1459 (8)	**/**	**/**	**194.21 ± 1.98**
36	2″-Pentosyl-6″-acetyl cytisoside ^c,^*	9.08	C_29_H_31_O_15_^−^	619.16684	619.17055	−3.70	101.0255 (18), 113.0246 (26), 131.0344 (12), 283.0643 (12), **307.0619 (100)**, 308.0676 (17), **325.0731 (62)**, 326.0740 (15), 337.0726 (44), 349.0726 (22), 367.0843 (33), 409.0936 (17), 427.1057 (16), 469.1162 (13), 619.1704 (16)	**/**	**/**	**35.15 ± 0.93**
**∑**								**/**	**/**	**950.92**
**Other detected flavonoids**										
37	Kaempferol-3-*O*-(6″-hexosyl)hexoside-7-*O*-hexoside with HCOOH ^c,^***	6.53	C_34_H_41_O_23_^−^	817.20390	817.20689	−2.99	**284.0375 (3)**, 285.0391 (4), 288.6129 (2), 299.0519 (2), 446.0897 (4), **447.0986 (48)**, 448.1022 (16), 489.0970 (3), 609.1533 (100), 610.1576 (39), 611.1627 (9), 612.1502 (2), 771.2062 (5), 772.2040 (3)	**/**	**6.50 ± 0.08**	**/**
38	Kaempferol-3-*O*-sinapoyl-trihexoside-7-*O*-hexoside ^c,^***	7.04	C_50_H_59_O_30_^−^	1139.30910	1139.32320	−14.10	**1139.318 (100)**, 815.2109 (52), **977.2654 (43)**, 609.1536 (9), **284.0357 (8)**	**/**	**26.27 ± 0.25**	**/**
39	Kaempferol-3-*O*-sinapoyl-dihexoside-7-*O*-hexoside ^c,^***	7.07	C_44_H_51_O_25_^+^	979.27190	979.27923	−7.33	127.0423 (4), **207.0669 (95)**, 208.0701 (16), 225.0765 (4), **287.0564 (55)**, 288.0608 (11), 291.0869 (3), 351.1114 (31), 352.1174 (7), **369.1216 (100)**, 370.1249 (26), 371.1262 (6), **449.1127 (49)**	**/**	**24.84 ± 0.72**	**/**
40	Quercetin 3-*O*-(6″-rhamnosyl)-hexoside ^c^	8.02	C_27_H_29_O_16_^−^	609.14611	609.14934	−3.24	151.0039 (3), 178.9985 (4), 255.0319 (3), 271.0253 (7), **300.0283 (100)**, 301.0355 (73)	**24.37 ± 0.36 ^A^**	**2.01 ± 0.02 ^C^**	**3.06 ± 0.07 ^B^**
**∑**								**24.37**	**59.63**	**3.06**
**∑∑**								**646.38**	**1588.71**	**1331.0**

Abbreviations: “/”—nonidentified phenolic compounds; AM—amaranth microgreens; BC—broccoli microgreens; RB—red beet microgreens; “*”—compounds detected only in negative ionization mode; “**”—compounds detected only in positive ionization mode; “***”—previously detected compounds in broccoli microgreens and reported by Liu and Shi [10]; “****”—apigenin *C* glycosides detected in accordance to previously reported date by Isayenkova and Wray [39] and da Silva and Morelli [40]. Compound quantities expressed using available standards ^a^; compounds expressed as sinapic acid equivalents ^b^; compounds expressed as apigenin equivalents ^c^. Means with the same uppercase letter in the same raw are significantly different according to Duncan’s test, (*p* < 0.05), (mean ± S.D.; *n* = 3).

**Table 3 foods-13-00757-t003:** Characterization and relative content (%) of betalains in amaranth and red beet microgreen samples by UHPLC Q-ToF-MS. Target compounds, expected retention time (RT), molecular formula, calculated mass, exact mass, and MS fragments are presented.

No	Compounds Name	RT	Formula	Calculated Mass	*m*/*z* Exact Mass	mDa	MS Fragments (% of Base Peaks)	Samples (%)		Ref
								AM	RB	
**Betalains**										
41	(Iso)Amaranthin	5.05	C_30_H_35_N_2_O_19_^+^	727.18285	727.18431	−1.45	150.0552(1), **389.0982 (100)**, 390.1014 (28), 391.1044 (5), **551.1509 (5)**, 552.1541 (2), 727.1837 (21)	**73.56**	**0.17**	[41,43,44]
42	(Iso)betanin	5.72	C_24_H_27_N_2_O_13_^+^	551.15130	551.15259	−1.29	150.0549 (2), 343.0931 (2), **389.0987 (100)**, 390.1021 (29), 391.1041 (5), 551.1522 (4)	**5.22**	**49.20**	
43	17-Decarboxy- (iso)amaranthin	5.85	C_29_H_35_N_2_O_17_^+^	683.19302	683.19430	−1.27	150.056 (1), **345.1084(100)**, 346.1116 (27), 347.1135 (4), **507.1618 (9)**, 508.1649 (4), 683.1930 (31)	**17.35**	**-**	
44	(2, 15 or 17)- Decarboxy- (iso)betanin	5.93	C_23_H_27_N_2_O_11_^+^	507.16150	507.16233	−0.83	106.0660 (2), 150.0549 (2), 299.1035 (1), 301.1186 (1), **345.1089 (100)**, 346.1124 (25), 347.1145 (4), 507.1617 (5)	**1.66**	**23.29**	
45	(2, 15 or 17)- Decarboxy(iso) betanin	6.40	C_23_H_27_N_2_O_11_^+^	507.16150	507.16406	−2.56	150.0548 (2), 299.1030 (2), 301.1176 (1), **345.1088 (100)**, 346.1126 (25), 347.1144 (4), 507.1622 (2)	**2.22**	**23.55**	
46	(2, 15 or 17)- Decarboxy-(iso) betanidin	7.01	C_17_H_17_N_2_O_6_^+^	345.10811	345.11267	−4.56	100.0392 (27), 106.0643 (37), 132.0449 (53), 144.0302 (47), **150.0541 (99)**, 151.0626 (62), 152.0708 (36), 202.0881 (34), 209.0726 (36), 227.0862 (35), 253.0849 (61), 255.1138 (65), 281.0767 (49), 299.1034 (43), **345.1061 (100)**	**-**	**3.79**	
**Total (%)**								**100**	**100**	

Abbreviations: AM—amaranth microgreens; RB—red beet microgreens.

**Table 4 foods-13-00757-t004:** Proximate compositions of broccoli, red beet, and amaranth microgreen juices.

Microgreen Juices	Family and Species	Yield of Juices (%)	Percentage of Dry Weight (%)	Percentage of Moisture (%)	pH Values	°Brix
BCJ	*Brassica oleracea* var. *italica*	70.20 ± 0.15 ^a^	1.84 ± 0.05 ^b^	98.26 ± 0.05 ^b^	5.96 ± 0.01 ^c^	2.00 ± 0.01 ^a^
RBJ	*Beta vulgaris*	62.00 ± 0.10 ^b^	2.00 ± 0.01 ^a^	98.00 ± 0.01 ^c^	6.44 ± 0.01 ^b^	2.00 ± 0.01 ^a^
AMJ	*Amaranthus tricolour* L.	53.40 ± 0.15 ^c^	1.64 ± 0.01 ^c^	98.46 ± 0.01 ^a^	6.52 ± 0.01 ^a^	1.80 ± 0.01 ^b^

Abbreviations: AMJ—cold-pressed amaranth microgreen juice; BCJ—cold-pressed broccoli microgreen juice; RBJ—cold-pressed red beet microgreen juice. Results were presented as mean values ± standard deviation. Different small letters in the same column denote a significant difference according to Duncan’s test, *p* < 0.05.

**Table 5 foods-13-00757-t005:** Characterization and quantification (mg/100 mL juices) of phenolic compounds detected in amaranth, red beet, and broccoli microgreen juices by UHPLC-QToF-MS. Target compounds, expected retention time (RT), molecular formula, calculated mass, exact mass, and MS fragments are presented.

No	Compounds Name	RT	Formula	Calculated Mass	*m*/*z* Exact Mass	mDa	MS Fragments (% of Base Peaks)	Samples (mg/100 mL)		
								AMJ	BCJ	RBJ
**Phenolic acid and derivatives**										
1a	Hydroxy- benzoic acid hexoside isomer I ^b^	1.68	C_13_H_15_O_8_^−^	299.07724	299.08054	−3.30	**137.0248 (100)**, 138.0301 (10)	**/**	**3.47 ± 0.03**	**/**
2a	Shikimic quinic acid hexoside ^b^	2.15	C_22_H_23_O_15_^−^	527.10370	527.11099	−7.29	143.0015 (3), 167.0342 (4), 173.0272 (2), **191.0555 (100)**, 192.0593 (10), 193.0580 (3), 353.0837 (4)	**0.92 ± 0.01**	**/**	**/**
3a	Dihydroxy- benzoic acid isomer I ^b^	2.35	C_7_H_5_O_4_^−^	153.01933	153.02154	−2.21	**108.0229 (100)**, 109.0294 (81), 110.0311 (6)	**/**	**0.72 ± 0.02**	**/**
4a	Hydroxy- benzoic acid dihexoside ^b^	3.46	C_24_H_19_O_10_^−^	467.09837	467.09935	−0.98	**137.0255 (100)**, 138.0284 (9), **299.0774 (2)**	**2.40 ± 0.02**	**/**	**/**
5a	Hydroxy- benzoic acid ^b^	3.91	C_7_H_5_O_3_^−^	137.02390	137.02614	−2.24	/	**14.97 ± 0.19 ^A^**	**5.15 ± 0.01 ^C^**	**9.81 ± 0.02 ^B^**
6a	Dihydroxy- benzoic acid isomer III ^b^	4.23	C_7_H_5_O_4_^−^	153.01933	153.02021	−0.88	106.9976 (65), 107.0293 (52), 107.053(20), **108.0218 (100)**, 122.9839 (14), 123.0203 (34), 135.0194 (11), 135.0538 (12)	**/**	**0.99 ± 0.03**	**/**
7a	Vanillic acid pentosyl hexoside ^b^	4.43	C_19_H_25_O_13_^−^	461.13007	461.13222	−2.16	108.0226 (5), 123.0461(7), 152.0122 (18), 153.0161 (3), **167.0374 (100)**, 168.0382 (10)	**6.86 ± 0.03**	**/**	**/**
8a	Dihydroxy- benzoic acid pentoside ^b^	4.71	C_12_H_13_O_8_^−^	285.06159	285.06303	−1.44	**108.0231 (100)**, 109.0291 (22), 110.0312 (2), 152.0117 (47), 153.0164 (9), 154.0176 (2)	**/**	**/**	**7.33 ± 0.03**
9a	Dihydroxy- benzoic acid pentosyl hexoside ^b^	5.32	C_18_H_23_O_13_^−^	447.11390	447.11985	−5.95	101.02230(3), 108.0229 (13), 109.0289 (14), 136.0394 (11), 151.0374 (3), **152.0114 (100)**, 153.0161 (14), 161.0453 (3), 163.0387 (6), 315.0666 (3), 447.1152 (46)	**13.83 ± 0.07**	**/**	**/**
10a	Benzoic acid derivative (like as carboxy benzoic acid) ^b^	5.66	C_8_H_5_O_4_^−^	165.01880	165.02126	−2.46	105.0153(52), 105.0395 (58), 108.0156 (13), 120.0197(38), **121.0306 (100)**, 122.0288 (11), 123.9880 (27), 124.0190(39), 135.0394 (17), 147.8908 (8), 151.9801(24), 152.0114 (32)	**/**	**48.02 ± 0.02**	**/**
11a	Hydroxy- benzoic acid hexoside isomer II ^b^	5.78	C_13_H_15_O_8_^−^	299.07724	299.08174	−4.50	**137.0252 (100)**, 138.0307 (9)	**/**	**/**	**1.06 ± 0.04**
12a	Coumaroyl- quinic acid isomer I ^b^	5.78	C_16_H_17_O_8_^−^	337.09289	337.09853	−5.64	111.0452 (5), **119.0519 (100)**, 120.0542 (11), 163.0406 (50), 164.0437 (7), 173.0448 (4), 191.0564 (60)	**/**	**55.84 ± 0.14**	**/**
13a	Benzoic acid ^b^	5.80	C_7_H_5_O_2_^−^	121.02900	121.03002	−1.02	/	**3.68 ± 0.02 ^B^**	**14.80 ± 0.02 ^A^**	**0.46 ± 0.01 ^C^**
14a	5-*O*-Caffeoyl- quinic acid isomer I ^b^	6.19	C_16_H_17_O_9_^−^	353.08781	353.08964	−1.83	135.0452 (1), 161.0242 (2), 173.0454 (1), **191.0554 (100)**	**/**	**1.61 ± 0.02**	**/**
15a	Coumaric acid hexoside ^b^	6.27	C_15_H_17_O_8_^−^	325.09230	325.09815	−5.85	117.0354 (6), **119.0513 (100)**, 120.0544 (11), **163.0398 (24)**, 164.0436 (3)	**2.63 ± 0.03**	**/**	**/**
16a	Carboxy hydroxybenzoic acid ^b^	6.53	C_8_H_5_O_5_^−^	181.01370	181.01425	−0.55	107.0304(15), 107.0612 (12), 117.0185 (9), **119.0235 (100)**, 120.0294 (16), 134.0376(30), 135.0487 (14), **137.0287 (26)**	**/**	**0.66 ± 0.01**	**/**
17a	Sinapic acid hexoside ^b^	6.59	C_17_H_21_O_10_^−^	385.11402	385.11652	−2.50	149.0249(21), 164.0481 (56), 165.0516 (8), 175.0042 (12), 179.0701(14), **190.0274 (100)**, 191.0325 (25), **205.0510 (99)**, 206.0569(45), 207.0492 (9), 217.0156 (11), 221.0806(14), 223.0620 (12)	**/**	**18.15 ± 0.02**	**/**
18a	Ferulic acid hexoside ^b^	6.67	C_16_H_19_O_9_^−^	355.10346	355.10148	1.98	111.0102(50), 112.0133 (16), 113.0147 (18), **134.0379 (100)**, 135.0424(12), 149.0610 (28), 150.0672 (3), 154.9760 (5), 155.0063 (6), 157.0035 (3), 178.0270 (62), 179.0308 (9), **193.0504 (37)**, 194.0542 (6)	**1.41 ± 0.01**	**/**	**/**
19a	Hydroxy- benzoyl malic acid ^b^	6.84	C_11_H_9_O_7_^−^	253.03480	253.03892	−4.12	102.9829 (2), 103.0087 (2), 114.0580 (2), **121.0305(100)**, 122.0332(10), 123.0058 (2), 123.0383(2), 130.0424 (2)	**2.02 ± 0.02**	**/**	**/**
20a	Coumaroyl- quinic acid isomer II ^b^	6.88	C_16_H_17_O_8_^−^	337.09289	337.09441	−1.52	109.0311(2), 111.0446 (18), 112.0467 (2), 119.0508 (44), 120.0531 (5), 137.0257 (11), 138.0320 (1), 155.0348 (6), **163.0402 (26**), 164.0441(4), **173.0455 (100)**, 174.0484(10), 191.0549 (3)	**/**	**1.16 ± 0.02**	**/**
21a	Sinapic acid ^a^	7.88	C_11_H_11_O_5_^−^	223.06120	223.06222	−1.03	105.0352(1), **121.0308 (100)**, 122.0339 (9), 134.0359 (1), 135.0456 (13), 136.0548 (1), 148.0172 (5), 149.0248 (50), 150.0277 (5), 163.0396 (13), 164.0469 (5), 165.0197 (27), 166.0219 (3), 193.0142(60), 194.0177 (7)	**/**	**60.00 ± 0.54**	**/**
22a	Sinapoyl malic acid ^b^	7.94	C_15_H_15_O_9_^−^	339.07216	339.07540	−3.24	115.0047 (47), 116.0085 (2), 117.0301 (1), 121.0313 (8), 132.0226 (2), **133.0156 (43)**, 134.0193 (2), 147.0462 (7), **149.0248 (100)**, 150.0291 (11), 164.0480 (86), 165.0519 (10), 179.0716 (2), 208.0385 (2), 223.0620 (7)	**/**	**134.18 ± 0.04**	**/**
23a	Benzoylmalic acid ^b^	7.94	C_11_H_9_O_6_^−^	237.03990	237.04514	−5.24	114.9839 (2), 115.0099 (3), **121.0310 (100)**, 122.0333 (10)	**/**	**2.09 ± 0.02**	**/**
24a	Dihydroxy- benzoic acid dihexoside ^b^	8.15^−^	C_21_H_19_O_13_^−^	479.08260	479.08873	−6.13	108.0228(20), 109.0346 (21), 137.0257 (61), **152.0122 (67)**, 153.0151(18), **435.0914 (100)**	**/**	**8.21 ± 0.08**	**/**
25a	Hydroxyferulic acid ^b^	11.25^−^	C_10_H_9_O_5_^−^	209.04500	209.04494	0.06	**105.0353 (100)**, 107.0146(58), 121.0291 (23), 123.0439 (44), 125.0253 (13), 131.0141 (16), 149.02305 (77), 150.0276(13), 151.0024 (70), 165.0555 (18), 167.0333 (19), 191.0347(10), **193.0143 (63)**, 209.0157 (12)	**/**	**2.53 ± 0.04**	**/**
**∑**								**48.74**	**357.59**	**18.66**
**Apigenin C-glycosides**										
26a	2″-Hexosyl vitexin ^c^	7.54	C_27_H_31_O_15_^+^	595.16630	595.17146	−5.16	271.0586(29), 283.0586 (17), 295.0580 (13), **313.0710 (100)**, 337.0691(18), 367.0794 (10), 379.0794 (8), 397.0905(29), 415.1016 (45), **433.1133 (88)**	**/**	**/**	**28.81 ± 0.05**
27a	2″-Pentosyl vitexin ^c^	7.61	C_26_H_29_O_14_^+^	565.15570	565.16052	−4.82	283.0596(16), **313.0724 (100)**, 337.0699 (17), 343.0802 (9), 367.0806 (16), 379.0806 (11), 397.0921 (41), 415.1041(74), **433.1144 (100)**	**/**	**/**	**34.45 ± 0.05**
28a	2″-Hexosyl-6″- acetyl vitexin ^c^	7.81	C_29_H_33_O_16_^+^	637.17690	637.18029	−3.39	283.0596(4), 295.0573 (4), **313.0710 (10)**, 337.0694 (3), 355.0789 (28), 367.0793 (3), 397.0898 (6), 415.1022 (10), 457.1121 (12), **475.1226 (100)**	**/**	**/**	**4.34 ± 0.04**
29a	2″-Hexosyl-6″- malonyl vitexin ^c^	7.95	C_30_H_33_O_18_^+^	681.16670	681.17004	−3.34	271.0583(18), 283.0590 (8), 295.0587 (21), **313.0712 (100)**, 337.0684 (13), 379.0797 (7), 439.1005 (9), 457.1118 (9), 475.1212(14), 483.0908 (8), 501.1012 (23), **519.1140 (60)**	**/**	**/**	**26.43 ± 0.05**
30a	2″-Hexosyl cytisoside ^c^	8.29	C_28_H_33_O_15_^+^	609.18190	609.18759	−5.69	285.0753(32), 297.0745 (16), 309.0745 (13), **327.0844 (100)**, 351.0852 (18), 381.0953 (8), 393.0952 (8), 411.1073 (28), 429.1186(50), **447.1272 (91)**	**/**	**/**	**48.94 ± 0.12**
31a	2″-Rhamnosyl cytisoside ^c^	8.35	C_28_H_33_O_14_^+^	593.18700	593.19311	−6.11	297.0738(12), **327.0850 (57)**, 351.0832 (12), 357.0946 (7), 381.0956 (11), 393.0956 (8), 411.1058 (34), 429.1168 (58), **447.1273 (100)**	**/**	**/**	**5.48 ± 0.06**
32a	2″-Pentosyl cytisoside ^c^	8.42	C_27_H_31_O_14_^+^	579.17140	579.17499	−3.59	297.0746(14), **327.0847 (100)**, 351.0851 (15), 357.0956 (9), 381.0957 (14), 393.0954 (9), 411.1078 (43), 429.1174(71), **447.1276 (95)**	**/**	**/**	**46.00 ± 0.23**
33a	2″-Hexosyl-6″- malonyl cytisoside ^c^	8.56	C_31_H_35_O_18_^+^	695.18230	695.19050	−8.20	285.0741(24), 297.0736 (9), 309.0747(21), **327.0859 (100)**, 351.0840 (12), 393.0956 (8), 453.1166 (10), 471.1282 (12), **489.1373 (17)**, 497.1060 (8), 515.1173 (27), **533.1284 (77)**	**/**	**/**	**17.90 ± 0.07**
34a	Cytisoside (3′-Methyl vitexin) ^c^	8.62	C_22_H_23_O_10_^+^	447.12910	447.13521	−6.11	135.0459 (8), 297.0737 (51), 309.0717 (8), **327.0846 (100)**, 337.1007 (14), 351.0832(22), 357.0948 (14), 365.1001 (10), 381.0924(11), 393.0937 (15), 411.1024 (31), 429.1197 (16)	**/**	**/**	**4.71 ± 0.05**
35a	2″-Hexosyl-6″- acetyl cytisoside ^c^	8.69	C_30_H_35_O_16_^+^	651.19250	651.19531	−2.81	297.0739(3), 309.0738 (3), **327.0852 (8)**, 351.0839 (2), 369.0966 (27), 393.095(2), 411.1064 (7), 429.1165 (9), 471.1271 (10), **489.1382 (100)**	**/**	**/**	**67.44 ± 0.34**
36a	2″-Hexuronyl-6″-acetyl cytisoside ^c^	8.76	C_30_H_33_O_17_^+^	665.17180	665.17943	−7.63	297.0742(10), 309.0736 (18), **327.0865 (100)**, 351.0839 (12), 453.1161(11), 459.1265 (11), 471.1254(12), 489.1366(10), 515.1169 (21), **533.1280 (49)**	**/**	**/**	**18.49 ± 0.08**
37a	6″-Acetyl cytisoside ^c^	9.30	C_24_H_25_O_11_^+^	489.13914	489.14587	−6.73	297.0740(33), 309.074 (21), **327.0846 (95)**, 351.0846 (13), **369.0954 (100)**, 381.0946(18), 393.0946 (15), 411.1052 (34), 429.1160(45), 471.1267 (16)	**/**	**/**	**9.39 ± 0.04**
38a	2″-Malonyl-6″- acetyl- cytisoside ^c^	9.57	C_27_H_27_O_14_^+^	575.14010	575.14610	−6.00	127.0370(7), 129.1006 (7), 297.0701 (12), 309.0736(51), **327.0849 (66)**, 351.0842 (19), **369.0949 (100)**, 375.0937 (8), 393.0966 (13), 453.1132(9), 471.1253 (9)	**/**	**/**	**1.40 ± 0.01**
39a	Apigenin ^a^	10.44	C_15_H_9_O_5_^−^	269.04500	269.05007	−5.07	136.9884(53), 139.0059 (53), 141.0708 (26), 143.0506(19), 167.0342 (31), 169.0656 (44), 171.0446(35), 179.0495 (19), 195.0448 (50), 197.0606(25), 223.0392 (52), 241.0492 (43), 251.0359(19), **269.0453 (100)**	**1.06 ± 0.02 ^B^**	**/**	**9.39 ± 0.05 ^A^**
**∑**								**1.06**	**/**	**323.17**
**Other flavonoids**										
40a	Kaempferol-3-*O*-sinapoyl- dihexoside-7-*O*-hexoside ^c^	6.85	C_44_H_49_O_25_^−^	977.25630	977.26510	−8.80	**815.2079 (100)**, 816.2111(54), 977.2594 (20), 609.1468 (13), **284.0332 (9)**, 446.085 (3)	**/**	**4.77 ± 0.07**	**/**
41a	Chalcan-flavan 3-ol dimer ^c^	7.68	C_27_H_31_O_14_^−^	579.17140	579.17180	−0.40	116.0382(13), 117.0445 (1), 125.0248 (10), 151.0035 (2), 167.0345 (28), 179.0413 (1), 201.1035 (3), 203.0823 (31), 204.0835 (4), **245.0924(100)**, 246.0951(19), 247.0961 (2), 271.0607 (4), **289.0706 (47)**	**/**	**/**	**0.19 ± 0.01**
42a	Europetin ^c^	9.37	C_16_H_11_O_8_^−^	331.04594	331.04602	−0.08	110.0017(37), 111.0082 (6), 121.0299(14), 137.9962(20), 139.0037 (26), 140.0085 (4), **165.9906 (100)**, 166.9962(24), 181.0143 (11), 193.9856 (6), 243.0284 (5), 271.0239 (8), 287.0173 (5), 316.0210 32), 317.0235 (7)	**0.04 ± 0.001**	**/**	**/**
**∑**								**0.04**	**4.77**	**0.19**
**∑∑**								**49.84**	**362.37**	**342.02**
**Other detected compounds**										
43a	Tuberonic acid	9.63	C_12_H_17_O_4_^−^	225.11270	225.11193	0.77	109.0414(11), 109.0694 (11), **110.0387(100)**, 111.0439(16), 123.0416 (14), 123.0707 (10), 135.0837(10), 136.0548 (36), 161.0720 (8), 161.1000(9), 163.1125 (21), 179.1085 (8), 181.1220(24), 207.1026 (88), 208.1046 (13)	**+**	**+**	**-**
44a	Methyl jasmonate	11.53	C_13_H_19_O_3_^−^	223.13340	223.13275	0.65	120.0274(27), 121.0237 (23), 123.1018 (28), 141.8757(21), 142.0382 (32), 142.0750 (25), **143.0676 (100)**, 143.1133 (57), 143.1413(26), 151.0361 (38), 168.8724 (26), 205.8271(26), 205.8642 (22), 214.9475 (21)	**+**	**+**	**+**

Abbreviations: AMJ—cold-pressed amaranth microgreen juice; BCJ—cold-pressed broccoli microgreen juice; RBJ—cold-pressed red beet microgreen juice. Compound quantities expressed using available standards ^a^; Compounds expressed as sinapic acid equivalents ^b^; compounds expressed as apigenin equivalents ^c^. “/”—nonidentified phenolic compounds; “+”—other detected compounds. Means with the same uppercase letter in the same raw are significantly different according to Duncan’s test, (*p* < 0.05), (mean ± S.D.; *n* = 3).

**Table 6 foods-13-00757-t006:** Characterization and relative content (%) of betalains in amaranth (AMJ) and red beet (RBJ) microgreen juices by UHPLC-QToF-MS. Target compounds, expected retention time (RT), molecular formula, calculated mass, exact mass, and MS fragments are presented.

No	Compounds Name	RT	Formula	Calculated Mass	*m*/*z* Exact Mass	mDa	MS Fragments (% of Base Peaks)	Samples (%)		Ref
								AMJ	RBJ	
**Betalains and betaxanthins**										
45a	Amaranthin	2.68	C_30_H_35_N_2_O_19_^+^	727.18340	727.18497	−1.57	**389.0989 (100)**, 551.1524 (7)	**78.22**	**-**	[41,43,44]
46a	Betalamic acid	3.77	C_9_H_10_NO_5_^+^	212.05590	212.05630	−0.40	102.0344 (3), 106.0293 (9), **120.0454 (100)**, 121.0475 (12), 122.0468 (2), 130.0292 (3), 138.0547 (1), 148.0389 (8), 149.0394 (1), 166.0469 (1)	**7.51**	**8.29**	
47a	γ-Aminobutyric acid-betaxanthin	3.16	C_12_H_15_N_2_O_6_^+^	283.09300	283.09369	−0.69	102.0341(2),116.0698 (4), 119.0361(3), **136.0610(100)**, 137.0632(9), 148.0400 (60), 149.0426 (8), 212.0448(2), 237.0866(3), 239.0570 (4), 248.0540 (2), 266.0677 (3), 283.0931 (33), 284.0950 (8)	**-**	**0.86**	
48a	Isoamaranhthin	4.91	C_30_H_35_N_2_O_19_^+^	727.18340	727.18378	−0.38	**389.0985 (100)**, 551.1515 (6)	**8.78**	**-**	
49a	Betanin	5.11	C_24_H_27_N_2_O_13_^+^	551.15130	551.15207	−0.77	150.0540 (2), 343.0909 (2), 345.1058 (1), **389.0978(100)**, 390.0990(32),551.1503 (5)	**3.91**	**70.54**	
50a	Isobetanin	5.59	C_24_H_27_N_2_O_13_^+^	551.15130	551.15703	−5.73	150.0531 (2), 343.0895 (1), **389.0959 (100)**, 390.0999 (25), 391.1010(5), 551.1478 (4), 552.1526 (2)	**-**	**7.49**	
51a	Decarboxy- dehydro- (iso)amaranthin	5.86	C_29_H_33_N_2_O_17_^+^	681.17790	681.18521	−7.31	297.0847 (8), 299.0987 (3), **343.0913 (100)**, 505.1446 (28)	**1.58**	**-**	
52a	(2 or 17)- Decarboxy(iso)- betanin	6.13	C_23_H_27_N_2_O_11_^+^	507.16150	507.16866	−7.16	150.0535 (2), 151.0613 (1), 299.0993 (2), **345.1059(100)**, 346.1116 (27), 347.1114 (4), 507.1575 (3)	**-**	**1.33**	
53a	(2 or 17)- Decarboxy- neobetanin	6.13	C_23_H_25_N_2_O_11_^+^	505.14580	505.15438	−8.58	253.0948 (3), 255.0956 (4), 269.0894 (5), 281.0887 (2), 297.0851 (20), 298.0888 (5), 299.0987 (3), **343.0899 (100)**, 344.0943 (24), 345.1067 (49), 346.1086 (8), 505.1434 (10)	**-**	**1.53**	
54a	Isoleucine- betaxanthin	6.94	C_15_H_21_N_2_O_6_^+^	325.14000	325.14250	−2.50	104.0494(12),106.0621 (17), 119.0612(10), 132.0511(14), 133.0753 (35), 147.0868 (14), 148.0480(12), 150.0540 (16), 173.0704 (14), 189.1365(35), **191.081 (100)**, 192.0843(17), 205.1273(11), 233.1240 (17), 325.1360 (10)	**-**	**2.88**	
55a	6′-*O*-Feruloyl- betanin	7.41	C_34_H_35_N_2_O_16_^+^	727.19870	727.20406	−5.36	**389.0975 (100)**	**-**	**7.07**	
**Total (%)**								**100**	**100**	

Abbreviations: AMJ—cold-pressed amaranth microgreen juice; RBJ—cold-pressed red beet microgreen juice.

## Data Availability

The authors declare that the data supporting the findings of this study are available within the paper. Should any raw data files be needed in another format, they are available from the corresponding author upon reasonable request.

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
