# Peer review of "Broccoli, Amaranth, and Red Beet Microgreen Juices: The Influence of Cold-Pressing on the Phytochemical Composition and the Antioxidant and Sensory Properties"

_foods, 2024, doi:10.3390/foods13050757_

Round 1

Reviewer 1 Report

Comments and Suggestions for Authors

Dear Author

Here I leave the comments for your consideration:

#Section 2.4

Detail the chemical composition of the SPE used.

  #Section 2.5

  Individual glucosinolates, phenolic compounds and betalains were identified based their monoisotopic mass, MS fragmentation and previously reported data in literature. Cite the literature consulted for each group of phytochemical compounds

 #Section 2.9.2.

 The evaluator did not compare the samples with each other but rated the sample according to individual sensory attributes and overall acceptability.

Why? How can to discern which is better if we can't have the possibility to compare among them? It is the wrong methodology and in consequence, also the results obtained

#Section  3.1

 Moreover, gluconapin and gluconasturtiin were detected in different extracts, depending on the extraction conditions and the extractant used.

Why? 

Do you use different extraction procedures to compare equal samples?

#  Table 2

Describe the methodology employed for the quantification. Stds?

  #Conclusions

I suggest expanding this section and in particular concluding whether when preparing the juices there are losses of which compounds or activities, biological or not.

Comments on the Quality of English Language

I consider that the quality of the language is acceptable for this type of scientific work.

Author Response

We thank the reviewer for recognizing the importance and scientific value of our research. Also, thank you very much for all the suggestions and the good will to improve the quality of the manuscript. All the suggestions were accepted and the corrections are marked in red font in the revised manuscript.

Reviewer 2 Report

Comments and Suggestions for Authors

The article concerns the effect of cold-pressing on the phytochemical composition, antioxidant and sensory properties of broccoli, amaranth and red beet juices. The article is well written, contains a large number of results that are discussed with findings of other researchers. I have only one objection:

- why did the authors choose the described extraction method (extraction with stirring on a mechanical shaker) to prepare the microgreens for chromatographic analysis.

- And how did the authors verify that the extraction was exhaustive?

Author Response

We thank the reviewer for recognising the importance and scientific value of our research. Thank you also very much for all your suggestions to improve the quality of the manuscript. All your suggestions have been accepted and explained below.

Reviewer 3 Report

Comments and Suggestions for Authors

Title of the manuscript: Broccoli, amaranth and red beet microgreens juices: the influence of cold-pressing on the phytochemical composition, antioxidant and sensory properties

The research topic is interesting. The literature review could be more extensive to include current research on juices in the context of sensory and antioxidant properties.

Some open comments/recommendations from Reviewer:

2.1. Microgreens sample

Is it possible to obtain more information about the research material? (e.g. year of production, maturity stage, growing conditions)

what was the yield of broccoli, amaranth and red beet microgreens juices?

2.9. Sensory properties of cold-pressed microgreens juices and 2.9.2. Consumer acceptance evaluation

There is no information related to the samples (juices) preparation and presentation for sensory evaluations (e.g. individual samples, amount, containers, coding, taste neutraliser, sample temperature). The conditions of sensory tests have not been given.

2.9.1. Overall quality evaluation

It is a pity that such a simple scale was used in the sensory evaluation. Please refer to ISO standards or books. Why were the coefficients used and what was the basis for the given attributes? Who assessed the samples for overall quality? How many assessors took part in the assessment? Were there any repetitions of sensory assessments/sessions?

2.9.2. Consumer acceptance evaluation

How many consumers took part in the assessment? How were consumers recruited? (any criteria?)

2.10. Statistical analysis

please explain: quality ranking?

Has a statistical analysis of consumer acceptance been carried out?

Author Response

We thank the reviewer for the recognizing the scientific value of our research. Thank you also very much for all suggestions and the good will to improve the quality of the manuscript. All suggestions are accepted and corrections are marked in blue font in the revised manuscript.

Round 2

Reviewer 3 Report

Comments and Suggestions for Authors

The authors responded to the comments and included the corrections in the manuscript. I accept the paper in its current form.